# DNA polymerase ζ deficiency causes impaired wound healing and stress-induced skin pigmentation

Sabine S Lange , Sarita Bhetawal , Shelley Reh, Katherine Leslie Powell, Donna F Kusewitt, Richard D Wood

**DNA polymerase ζ (pol ζ) is well established as a specialized enzyme important for DNA damage tolerance, facilitating DNA synthesis past lesions caused by radiation or chemical damage. We report that disruption of *Rev3l* (encoding the catalytic subunit of pol ζ) in mouse epidermis leads to a defect in proliferation that impairs cutaneous wound healing. A striking increase in epidermal skin pigmentation accompanied both wound healing and UV irradiation in these mice. This was a consequence of stress-induced migration of *Rev3l*-proficient melanocytes to the *Rev3l*-defective epidermis. We found that this pigmentation corresponded with p53 activation in keratinocytes and was absent in p53-negative areas of the epidermis. Expression of the kit ligand (*Kitl*) gene, a p53-controlled mediator of keratinocyte to melanocyte signaling, was enhanced during wound healing or following UV irradiation. This study extends the function of pol ζ to the process of proliferation during wound healing. *Rev3l*-deficient epidermis may be a useful mouse model system for examining communication between damaged keratinocytes and melanocytes, including signaling relevant to human disease.**

## Introduction

Many types of DNA damage are impediments to DNA replication (1). One mode of DNA damage tolerance uses specialized DNA polymerases to bypass DNA damage (2). Arguably, the most critical of these specialized enzymes is DNA polymerase ζ (pol ζ). Evidence is accumulating that pol ζ is frequently used in response to circumstances that stall DNA replication forks. Such circumstances include encounters with template DNA damage or with DNA sequences that interrupt replication, such as chromosomal fragile sites (3, 4) and stable non-B DNA structures (5, 6). In the absence of *Rev3l*, which encodes the catalytic subunit of pol ζ, cells accumulate DNA double-strand breaks and chromosome aberrations (7, 8). Inactivation of *Rev3l* results in embryonic lethality in mice (9, 10).

In cells from adult animals, elimination of pol ζ function has varying consequences, depending on the target tissue. Disruption of *Rev3l* in

the hematopoietic system is not well tolerated (11, 12, 13). Mice are viable, however, after ablation of *Rev3l* in epithelial tissues using Cre recombinase driven by a keratin-5 promoter (14). These mice have substantially increased baseline levels of DNA damage stress, and their keratinocytes are extremely sensitive to killing by UV radiation (14).

Cells can proliferate in the absence of pol ζ in some circumstances. For example, T antigen–immortalized MEFs have a loss of checkpoint controls and can proliferate without functional *Rev3l*, but at the expense of an order of magnitude increase in frequency of DNA double-strand breaks and chromosome rearrangements (9, 14). Similarly, some human cancer cell lines can propagate following inactivation of *REV3L*, but with increased sensitivity to many DNA-damaging agents (15). One consequence of the genome protective function of pol ζ is that *Rev3l* is a suppressor of spontaneous tumor formation. An increased incidence and reduced latency of lymphomas was observed in *Tp53*-defective mice with mosaic deletion of *Rev3l* in the lymphocyte lineage (11). Conditional partial deletion of *Rev3l* in the mammary gland enhances the incidence of mammary gland tumorigenesis (11). Deletion of *Rev3l* in keratin-5–expressing cells results in spontaneous squamous cell carcinomas in ~90% of mice (14).

These experiments suggest that pol ζ may be necessary for continued proliferation of primary cells, even in the absence of exogenous DNA damage. We, therefore, examined wound healing, a process dependent on rapid proliferation responses. We report here the discovery that *Rev3l*-defective epidermis exhibits a proliferative defect in wound healing. Furthermore, we found that both wound healing and UV irradiation in adult mice engender striking epidermal skin pigmentation, caused by stress-induced migration of *Rev3l*-proficient melanocytes to the *Rev3l*-defective epidermis. This proliferation-associated pigmentation is p53 dependent and is absent in p53-defective areas of the epidermis. This study extends the function of pol ζ to the process of proliferation during wound healing. The results suggest *Rev3l*-deleting mice as a tool for dissecting communication between damaged keratinocytes and melanocytes, including signaling relevant to human disease.

Department of Epigenetics and Molecular Carcinogenesis, The University of Texas MD Anderson Cancer Center, and Graduate School of Biomedical Sciences at Houston, Smithville, TX, USA

Correspondence: rwood@mdanderson.org
Sabine S Lange's present address is Texas Commission on Environmental Quality, Austin, TX, USA.

# Results

## Abnormal wound architecture and proliferation defects in *Rev3l*-deficient skin

To determine whether a defect in pol ζ affects adult cellular proliferation in the absence of externally applied DNA damage, we carried out in vivo and ex vivo wound healing experiments. We used adult *Rev3l*[−/lox] mice carrying one disrupted *Rev3l* allele and one functional allele with essential residues flanked by *loxP* sites for targeting by Cre recombinase (Fig 1A). Control mice were *Rev3l*[+/lox], so that one normal *Rev3l* copy remained after disruption of the floxed allele. As an indicator of Cre recombinase action, the mice also harbored the mT/mG transgene (16). Constitutively expressed RFP is deleted upon expression of Cre, allowing expression of GFP (Fig 1B). Cre recombinase was driven by the bovine keratin-5 (BK5) promoter which is active in epithelia (17) but leaves *Rev3l* unaffected in the dermis (Fig 1C), melanocytes, and other non-epithelial tissues (14).

Excisional wounds were introduced into the dorsal skin of experimental and control mice, and wound re-epithelialization was followed through the steps of cell migration, proliferation, and differentiation. Defects in migration, but not proliferation, are associated with chronic nonhealing wounds (18). GFP fluorescence of the dorsal skin was monitored noninvasively daily (Fig 1D) to measure closure of the wound, which took place at the same rate in *Rev3l* deleting and proficient epidermis (Fig 1E). To examine proliferation and migration more closely, the skin of BK5.Cre; *Rev3l*[−/lox] and *Rev3l*[+/lox] mice was examined at day 3 of wound healing (Fig 1F). Migration of keratinocytes in the skin adjacent to the wound occurred in animals from both genotypes, as shown by staining with cytokeratin-6 (K6), a marker of migrating epithelial cells (19). In the epidermis immediately adjacent to the 3-d-old wound, DNA synthesis as measured by incorporation of bromodeoxyuridine (BrdU) was similar in the two genotypes (Fig 1G). However, epidermal cells that had migrated into the wound area showed markedly reduced proliferation in *Rev3l*-deficient skin, with 50% of the control level of BrdU incorporation (Fig 1G).

The architecture of the closed wound (day 10) was markedly abnormal in BK5.Cre; *Rev3l*[−/lox] mice. The basal epidermis was discontinuous and disorganized, contrasting with the organized stratification seen in BK5.Cre; *Rev3l*[+/lox] skin (Fig 2A). Staining with cytokeratin-10 (K10), which marks the suprabasal but not the basal

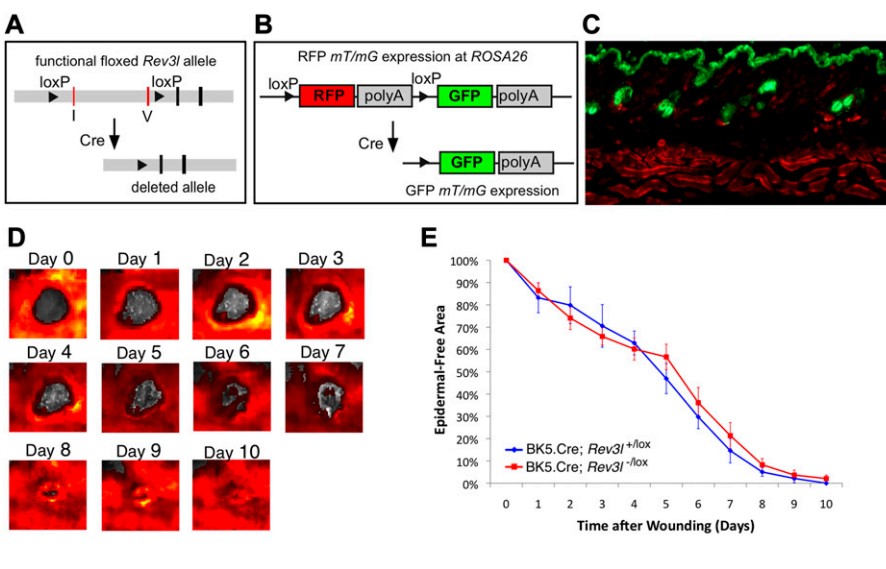

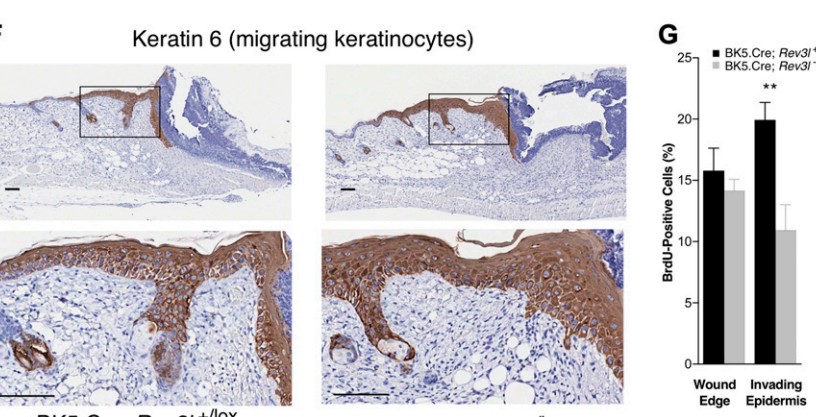

**Figure 1. Wounds in *Rev3l*-deficient epidermis can close by migration of non-proliferating epidermal cells.**
**(A)** LoxP sites flank exons encoding part of *Rev3l* DNA polymerase motif I and all of motif V necessary for polymerase activity (exons 28 and 29 of mouse Rev3l transcript variant NM_002912). **(B)** The mT/mG reporter gene in the mice switches fluorescence from red to green upon expression of Cre. **(C)** Mice expressing K5.Cre show loxP-mediated recombination (marked by GFP expression) in epidermal cells, but not dermal or panniculus carnosus cells. **(D)** IVIS spectrum pictures of GFP fluorescence (shown in red) and the wound site (shown in gray) over time after wounding (from day 0 to day 10) from one mT/mG transgene–expressing mouse. **(E)** Quantification of wound re-epithelialization of excisional wounds in BK5.Cre; *Rev3l*[+/lox] and BK5.Cre; *Rev3l*[−/lox] mice (n = 5 mice of each genotype) over a 10-d period after wounding. **(F)** Keratin-6 staining of skin sections 3 d after wounding. Distance bar: 100 μm. **(G)** Quantification of BrdU incorporation into epidermal cells 3 d after wounding, within 1 mm of the wound edge and in epidermal cells invading into the wound area (n = 5 mice of each genotype). Data points are mean ± SEM; *P < 0.05, **P < 0.01.

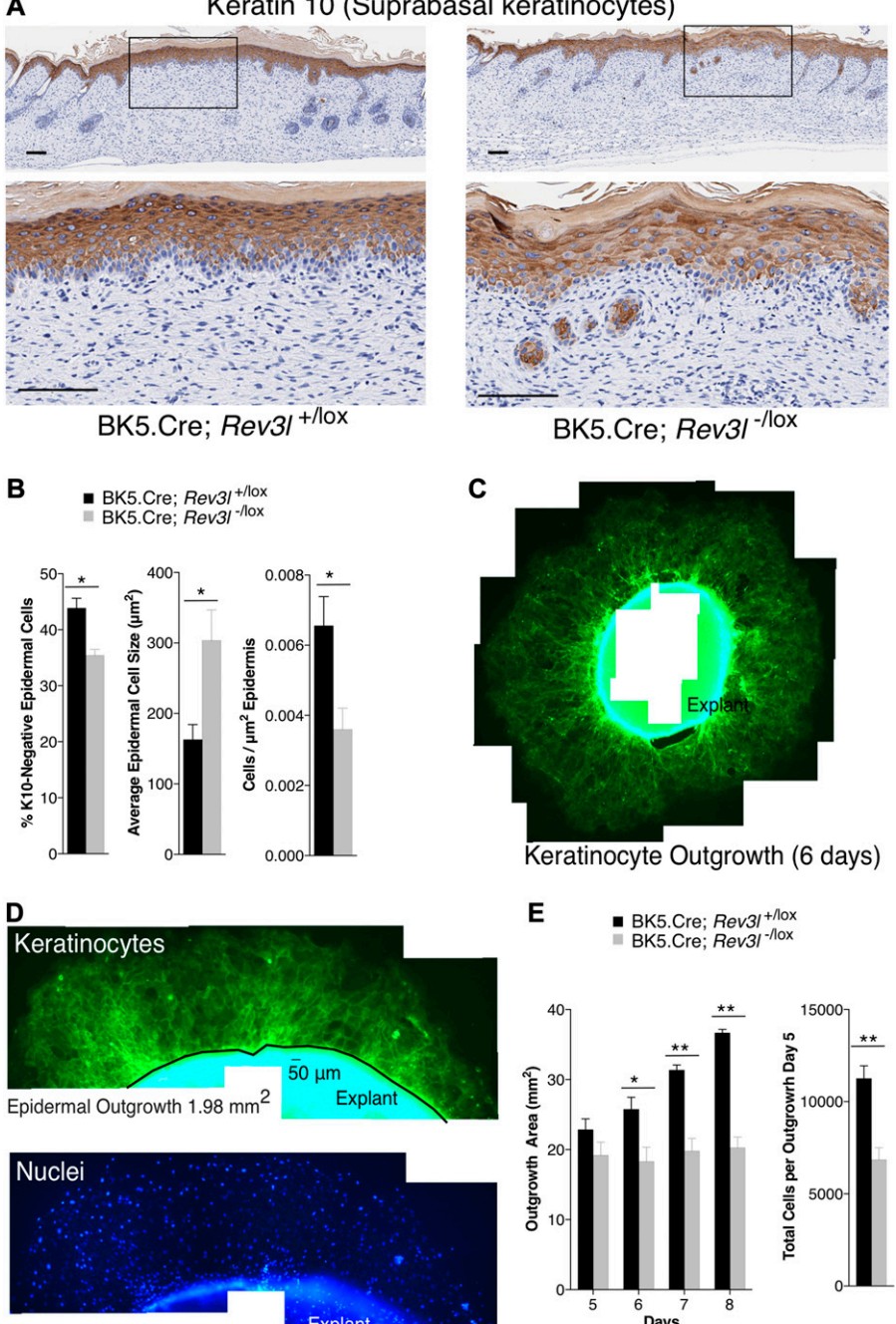

**Figure 2. Aberrant wound architecture and proliferation in *Rev3l*-deficient skin.**
**(A)** Keratin-10 staining of skin sections from wounds from BK5.Cre; *Rev3l*⁺/lox and BK5.Cre; *Rev3l*⁻/lox mice 10 d after wounding. Distance bar: 100 μm. **(B)** Quantification of keratin-10–negative epidermal cells (basal cells), average size of epidermal keratinocytes, and epidermal cell density (n = 4 mice of each genotype) 10 d after wounding. **(C)** GFP fluorescence of the keratinocyte outgrowth from a skin explant 6 d after plating. **(D)** GFP (keratinocyte) and Hoechst 33342 dye (nuclei) from a skin explant 5 d after plating in culture. The total area of cell outgrowth and the number of nuclei in that area of outgrowth are shown. **(E)** Quantification of the total outgrowth area from skin explants at 5–8 d after plating and cell density at 5 d after plating (n = 3 mice per genotype). Data points are mean ± SEM; *$P < 0.05$, **$P < 0.01$.

cells of the epidermis showed that there were fewer, larger cells in *Rev3l*-defective epidermis (Fig 2B). We extended the analysis by analyzing skin explants from BK5.Cre; *Rev3l*⁻/lox and BK5.Cre; *Rev3l*⁺/lox mice. GFP-positive (*Rev3l*-deficient) keratinocytes arising from isolated explants were monitored and measured over time after plating into culture (Fig 2C). Cell numbers were quantified by staining nuclei with Hoechst 33342 dye (Fig 2D). Emergence of cells from the *Rev3l*-deficient explants stopped at day 5, whereas outgrowth from the *Rev3l*-proficient explants continued through day 8 (Fig 2E). The average outgrowth area for all the skin explants was equal for

the two genotypes up to day 5, but *Rev3l*-deficient outgrowths contained only about half the number of keratinocytes as controls (Fig 2E). In summary, wound healing in *Rev3l*-deficient epidermis is aberrant because of a marked decrease in proliferation of keratinocytes.

## Abnormal epidermal pigmentation associated with stress stimuli in *Rev3l*-deleting skin

Melanocytes normally reside in the hair follicles of adult mice and export melanin to hair-producing keratinocytes (20). Unlike

human skin, normal-haired mouse skin remains unpigmented (constitutively or in response to UV radiation) because melanocytes are absent from the interfollicular epidermis (21, 22). Unexpectedly, we found abundant melanin pigment in the basal epidermis of *Rev3l*-deleting skin after wounding (Figs 3A and S1). To analyze the stimuli leading to pigmentation, BK5.Cre; *Rev3l*$^{-/lox}$ and *Rev3l*$^{+/lox}$ C57BL/6 mice were exposed to UVB radiation. Mice with intact *Rev3l* in the epidermis showed no UV-induced pigmentation (as expected), but melanin was deposited abundantly in the basal epidermis of *Rev3l*-deficient UV-irradiated mice (Figs 3B and S1). We also treated mice with 12-O-tetradecanoylphorbol-13-acetate (TPA), a phorbol ester that promotes cell proliferation and inflammation via the protein kinase C pathway (23). TPA promoted proliferation in *Rev3l*-proficient mouse skin but had only a minor effect on proliferation in *Rev3l*-deficient epidermis (14). We found that treatment of mice with TPA for 2 wk also caused pronounced pigmentation of *Rev3l*-deleted skin, with melanin observed in the epidermis of these mice (Figs 3C and S1).

Analysis of the time course of UV radiation–induced pigmentation showed that pigmentation appeared by 4 d after exposure (Fig 4). Maximal pigmentation was seen at 6 d and persisted for at least 10 d (Fig 4). Occasional patches of non-pigmented skin (Fig 4, arrow) appeared over time following UV irradiation in the *Rev3l*-deficient mice, as discussed below. Therefore, multiple stressors cause an abnormal epidermal pigmentation response in *Rev3l*-deficient skin.

## Melanocytes migrate to the epidermis of stressed *Rev3l*-deficient skin

To determine whether melanocytes are present in the epidermis of the stressed pigmented skin of BK5.Cre; *Rev3l*$^{-/lox}$ mice, we stained for TRP2 (product of the gene *Dct*), an enzyme involved in melanin synthesis that is specific for detection of differentiated melanocytes and melanocyte stem cells (20, 24, 25). To avoid interference from melanin, we examined skin from previously described non-pigmented albino mice that had been treated with TPA or with UV radiation (14). TRP2-positive cells were very rare in unchallenged *Rev3l*-proficient or *Rev3l*-deficient epidermis. TRP2-positive cells were present in the basal layer of *Rev3l*-deficient epidermis 3 d after UV irradiation (Fig 5A and D), after 2 or 40 wk of TPA application (Fig 5B and E), or 3 d after wounding (Fig 5C and F). These melanocytes apparently originated from hair follicles, where they are normally resident in mice (Fig S2).

## P53 stabilization in *Rev3l*-deficient keratinocytes

Proliferation in the absence of *Rev3l* causes accumulation of breaks in genomic DNA, which activate p53 and other DNA damage response factors (7, 14). In the BK5.Cre-driven model used here, this activation of DNA damage signaling occurs in wounded, UV-irradiated, or TPA-treated keratinocytes. The melanocytes are *Rev3l*-proficient in these mice (they do not express BK5.Cre). Therefore, it is important to gain insight into the signaling from *Rev3l*-defective

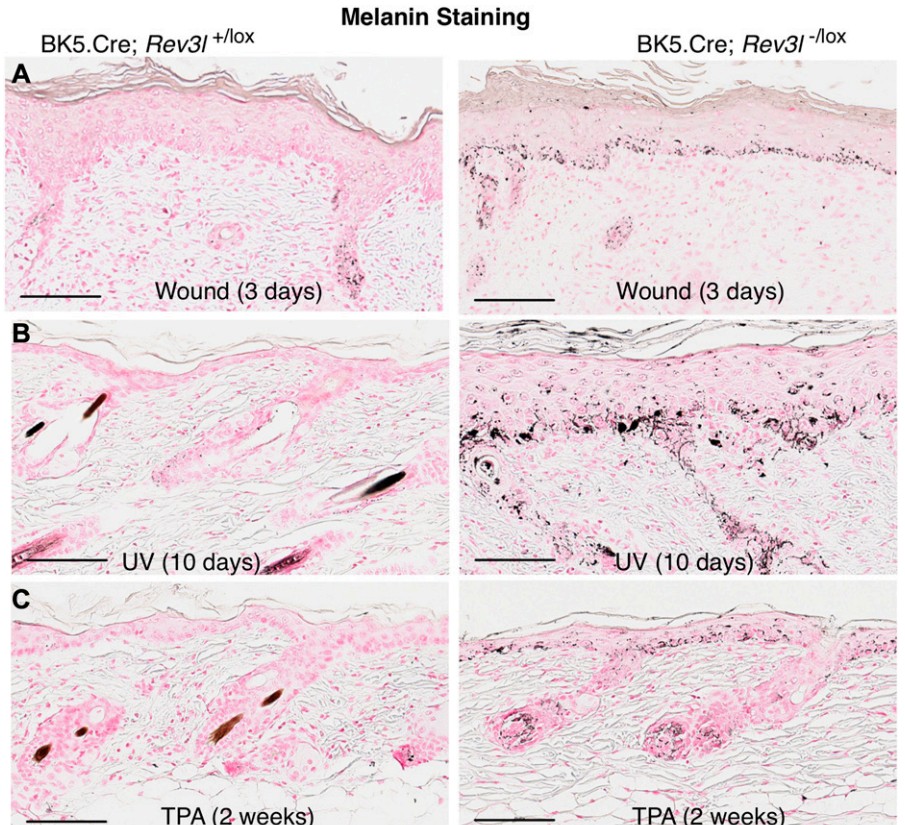

**Melanin Staining**

BK5.Cre; *Rev3l* $^{+/lox}$

BK5.Cre; *Rev3l* $^{-/lox}$

A — Wound (3 days)

B — UV (10 days)

C — TPA (2 weeks)

**Figure 3. Pigmentation in *Rev3l*-deleted interfollicular epidermis with wounding, UV radiation, or TPA application.**
Fontana-Masson staining of skin sections from BK5.Cre; *Rev3l*$^{+/lox}$ and BK5.Cre; *Rev3l*$^{-/lox}$ mice **(A)** 3 d after wounding, **(B)** 10 d after UV radiation, or **(C)** 2 wk after TPA application. Melanin pigment stains black. Distance bar: 100 μm.

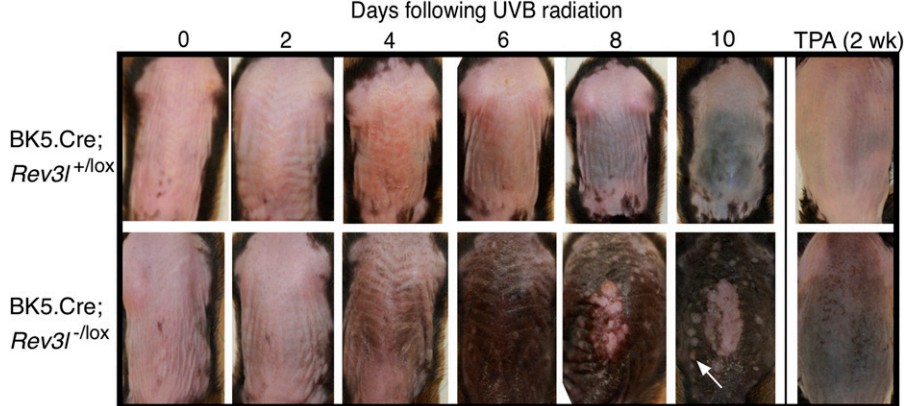

**Figure 4. Abundant interfollicular pigmentation following UV radiation of mice with *Rev3l*-deficient epidermis.**
Time course of pigmentation following irradiation of mice with one functional copy of *Rev3l* (top row) or with inactivation of both copies in K5-expressing keratinocytes (bottom row). The white arrow points to an example of a non-pigmented area. At day 10, *Rev3l*-proficient mice began to regrow hair, resulting in the mildly darkened appearance. The right-most panel shows evidence of epidermal pigmentation after 2 wk of TPA application.

keratinocytes that triggers an unusual response in the *Rev3l*-proficient melanocytes.

Because melanocytes are normally resident in the hair follicles, we measured the proportion of hair follicles in undamaged skin that had p53-positive cells in the bulge region (which contains keratinocyte and melanocyte stem cells) or in the hair bulb. In *Rev3l*-proficient hair follicles, only weak p53 staining was observed in fewer than 30% of the bulge area cells and less than 20% of the hair bulbs. In contrast, hair follicles in *Rev3l*-deficient skin had very strong p53 staining in almost 100% of the bulge and hair bulb regions (Fig 6A and B). In addition to the stabilization of p53, the increased DNA double-strand breaks in proliferating pol ζ–defective cells were detected by γ-H2AX formation (14). We found γ-H2AX–positive cells in similar regions of the *Rev3l*-deficient hair follicles (Fig 6C and D).

Adult mice have epidermal melanocytes only in the feet, the tail, and the ears. Several mouse models have been described with constitutively activated p53 arising from endogenous stress, and these animals have more darkly pigmented feet, referred to as the "sooty foot" pattern (26, 27, 28). We, therefore, examined foot pigmentation in the BK5.Cre-deleting mice. Indeed, darker pigmentation was observed in the footpad of animals with both *Rev3l* alleles inactivated in epidermal keratinocytes (Fig S3). This increased melanin production in a location that naturally harbors melanocytes in the epidermis is consistent with constitutive activation of p53 in *Rev3l*-defective epidermis.

Keratinocytes secrete signaling molecules that affect the proliferation, migration, and melanin production of melanocytes (25). These include KITL, pro-opiomelanocortin-alpha (POMC, also known as alpha-melanocyte stimulating hormone), and endothelin 1 (EDN1), all of which have gene transcripts directly regulated by p53 (27, 29, 30); transcription of human transforming growth factor, beta 1 (TGFB1) and WNT signaling pathway genes can be indirectly affected by p53 (31, 32). We isolated RNA from the epidermis and hair follicle keratinocytes of untreated, *Rev3l*-deficient, and *Rev3l*-proficient skin from pigmented mice and measured the expression levels of these genes. The RT–PCR confirmed the absence of *Rev3l* expression in keratinocytes from the BK5.Cre; *Rev3l*⁻/ˡᵒˣ mice (Fig 6E). We found an increase in *Kitl* expression in the *Rev3l*-deficient compared with proficient keratinocytes (Fig 6F) and a possible increase in *Edn1* (Fig 6G). No change was observed in *Tgfb1* levels

(Fig 6H), whereas little *Wnt1* or *Pomc* signal was detected. The enhanced expression of the melanocyte-regulating gene *Kitl* in the untreated keratinocytes of *Rev3l*-deficient mice is likely to contribute to the increased signaling to melanocytes.

## P53 and γ-H2AX accumulate in stressed *Rev3l*-deficient epidermis and hair follicles

When present in the epidermis, melanocytes deposit pigment in response to signals from epidermal keratinocytes. The pigmentation response that follows UV radiation of the skin of *Rev3l*-deficient mice is preceded by increased γ-H2AX staining (reflecting DNA damage) and highly activated p53 in epidermal and hair follicle keratinocytes (14). By immunohistochemical staining, we found that wounding, like UV radiation, increased the number of γ-H2AX– and p53-positive cells in *Rev3l*-deficient mice compared with *Rev3l*-proficient mice in both the interfollicular epidermis (Fig 7) and the hair follicles (Fig S4).

The data are consistent with a mechanism whereby persistent strong signaling from keratinocytes causes the activation and migration of melanocytes from hair follicles to epidermis and deposition of melanin. When melanocytes are already resident in the epidermis, as in the mouse footpad, inactivation of *Rev3l* in keratinocytes is sufficient to promote melanization. The baseline level of DNA damage stress without additional radiation or wounding is not sufficient to cause melanocyte migration and melanin deposition in the interfollicular epidermis. However, in mice with *Rev3l*-defective epidermal keratinocytes, a very strong signal is caused by damage and proliferation induction following UV radiation that drives migration of melanocytes from hair follicles and deposition of melanin in the epidermis. An intermediate but pronounced signal is provoked by wound healing of the interfollicular epidermis.

TPA caused an increase in γ-H2AX formation, proliferation, DNA breaks, and p53 stabilization in the hair follicles, but not in the epidermis of *Rev3l*-deficient skin (Figs 7 and S4). *Rev3l*-deficient epidermis has innately poor proliferative capacity, which is likely why TPA does not stimulate significant proliferation (14) or increase DNA breaks and p53 stabilization. The mechanism of TPA as a pro-pigmentation signal in *Rev3l*-deficient skin may involve known activities such as enhancement of cleavage of KITL to its soluble, signaling form (33) and activation of the protein kinase C pathway (34).

## Melanocytes (TRP2 staining)

BK5.Cre; *Rev3l* $^{+/lox}$    BK5.Cre; *Rev3l* $^{-/lox}$

**A** 4 days UV

**B** 2 weeks TPA

**C** 3 days Wound

**D**

TRP2-Positive / mm skin

■ BK5.Cre; *Rev3l* $^{+/lox}$
▬ BK5.Cre; *Rev3l* $^{-/lox}$

Days After UV Radiation

**E**

TRP2-Positive / mm skin

Weeks of TPA

**F**

TRP2-Positive / mm skin

3 days Wound

## Melanocytes are not recruited to areas that lack p53 activation in *Rev3l*-deficient skin

The experiments summarized above suggest that the signaling cascade causing melanocytes to proliferate, migrate, and produce melanin is initiated by p53 stabilization in response to DNA breaks. If so, cell clones without p53 activation would not signal to melanocytes.

We observed areas of *Rev3l*-deficient epidermis that showed no p53 stabilization after UV irradiation. These p53-unresponsive areas provided an opportunity to test the role of p53 in melanocyte activation. Patches of skin with and without p53 accumulation after UV radiation were analyzed (Fig 8A). The p53-negative areas proliferated following UV irradiation (Fig 8B), even while having high levels of DNA breaks as demonstrated by γ-H2AX staining (Fig 8C). This lack of p53 activation allows damaged cells to escape cell death and proliferate.

*Rev3l*-deficient basal epidermis with low proliferation and high p53 activation was populated with melanocytes (TRP2-positive cells) (Fig 8D). However, areas containing epidermal keratinocytes with high proliferation (BrdU incorporation) and low or no stabilized p53 had very few melanocytes (Fig 8D). This may account for the non-pigmented areas in the *Rev3l*-deficient skin that become more apparent with time after UV irradiation (Fig 4).

# Discussion

### DNA pol ζ is required for normal wound healing

This study tested whether wound healing, a response that requires epidermal cell proliferation, is dependent on pol ζ function. We found that wound healing was defective in *Rev3l*-deficient mouse epidermis, even in the absence of externally applied DNA damage. *Rev3l*-defective keratinocytes could migrate, allowing an initial epidermal wound closure step. However, their pronounced proliferative defect once resident in the wound bed led to architecturally aberrant, disordered wound healing. This included deficits in the basal layer of the epidermis, hypocellularity, and the presence of enlarged suprabasal cells. Consistent with this, skin explants of *Rev3l*-deficient keratinocytes were not able to sustain significant outgrowth. Although migration of keratinocytes from explants did occur, the cells within the outgrowth were unable to proliferate for more than a few days.

The results support the interpretation that pol ζ inactivation is incompatible with the rapid proliferation of normal mammalian cells with intact cell cycle checkpoints. For example, the rapidly dividing and developing mouse embryo loses viability during gestation in the absence of pol ζ. A broad inactivation of *Rev3l* in hematopoietic progenitors in the adult also does not give rise to propagating B or T cells (11), although disruption is tolerated in

mature resting B cells expressing CD19 or CD21 (12, 13). The experiments described here were possible because mice can survive elimination of *Rev3l* function in epidermal keratinocytes. However, these animals display baseline skin abnormalities, including reduced epidermal cellularity and irregular hair cycling, with gradual hair loss (14).

### Hyperactivation of a p53-dependent pigmentation pathway in pol ζ–defective epithelium

We discovered an unexpected consequence of proliferation in the absence of pol ζ: stress-induced epidermal pigmentation. Pigmentation after UV radiation of human skin protects the cell nucleus against additional DNA damage by absorbing further radiation (20). In mice, UV radiation–induced tanning does not normally occur in the bulk of the epidermis. One major reason for this is that, in contrast to human skin, melanocytes are not resident throughout the epidermis (20, 21). Instead, melanocytes in the mouse are largely localized in the hair follicles. Exceptions are the feet, the tail, and the pinnae of the ear, which do harbor epidermal melanocytes that pigment the epidermis upon DNA damage stimulation in a p53-dependent manner (29). Several genetic models or developmental stages can alter melanocyte distribution and the propensity for stress-induced pigmentation. The skin of murine neonates (<10 d old) contains widespread epidermal melanocytes that respond to UV radiation or wounding by proliferating and pigmenting the epidermis (22, 35). Activation of KITL, for example, by expression in keratinocytes with a K14 promoter, leads to localization of melanocytes throughout the epidermis (36, 37). Pigmentation by melanin deposition further requires signaling to melanocytes by POMC, which activates melanocytes via cAMP (37). Both the KITL and POMC pathways appear necessary for skin pigmentation in adult mice.

Several mouse models with increased p53 stabilization have darker pigmentation in the non-haired skin of mice, particularly the feet. These include models where p53 is stabilized by a decrease in the p53-negative regulator MDM2 or where cellular stress is elevated by impairing ribosomal function (26, 27, 28, 29, 30). Mice with a catalytically-dead DNA protein kinase subunit have impaired DNA break repair and also show p53-dependent pigmentation of non-haired skin (38). In these cases, the stabilization of p53 in epidermal areas that already have resident melanocytes is enough to stimulate those melanocytes to produce pigment. In these models, however, there is no migration of melanocytes to the epidermis of unchallenged haired skin. We find here that mice with *Rev3l*-deficient epidermal keratinocytes have a baseline level of DNA damage stress that is sufficient to activate p53 and initiate pigmentation in non-haired skin by resident epidermal melanocytes (Fig 9).

*Rev3l*-deficient hair follicles contain melanocytes that respond rapidly to epidermal stress. Persistent DNA damage in the hair

---

**Figure 5. Melanocytes in *Rev3l*-deficient epidermis following UV radiation, wounding, or TPA.**
TRP2 staining (marking melanocytes) of skin sections from BK5.Cre; *Rev3l*⁺/lox and BK5.Cre; *Rev3l*⁻/lox mice **(A)** 4 d after UV radiation, **(B)** 2 wk after TPA application, or **(C)** 3 d after wounding; arrows point to examples of TRP2-positive melanocytes. Quantification of epidermal TRP2-positive cells per millimeter of skin is shown for **(D)** no UV or 1–10 d after UV radiation (n = 3 mice of each genotype), **(E)** 2 wk after acetone (ctrl) or TPA application or 40 wk after TPA application (n = 4 mice of each genotype), or **(F)** 3 d after wounding (n = 5 mice of each genotype). Data points are mean ± SEM; **P < 0.01. Distance bar = 100 μm.

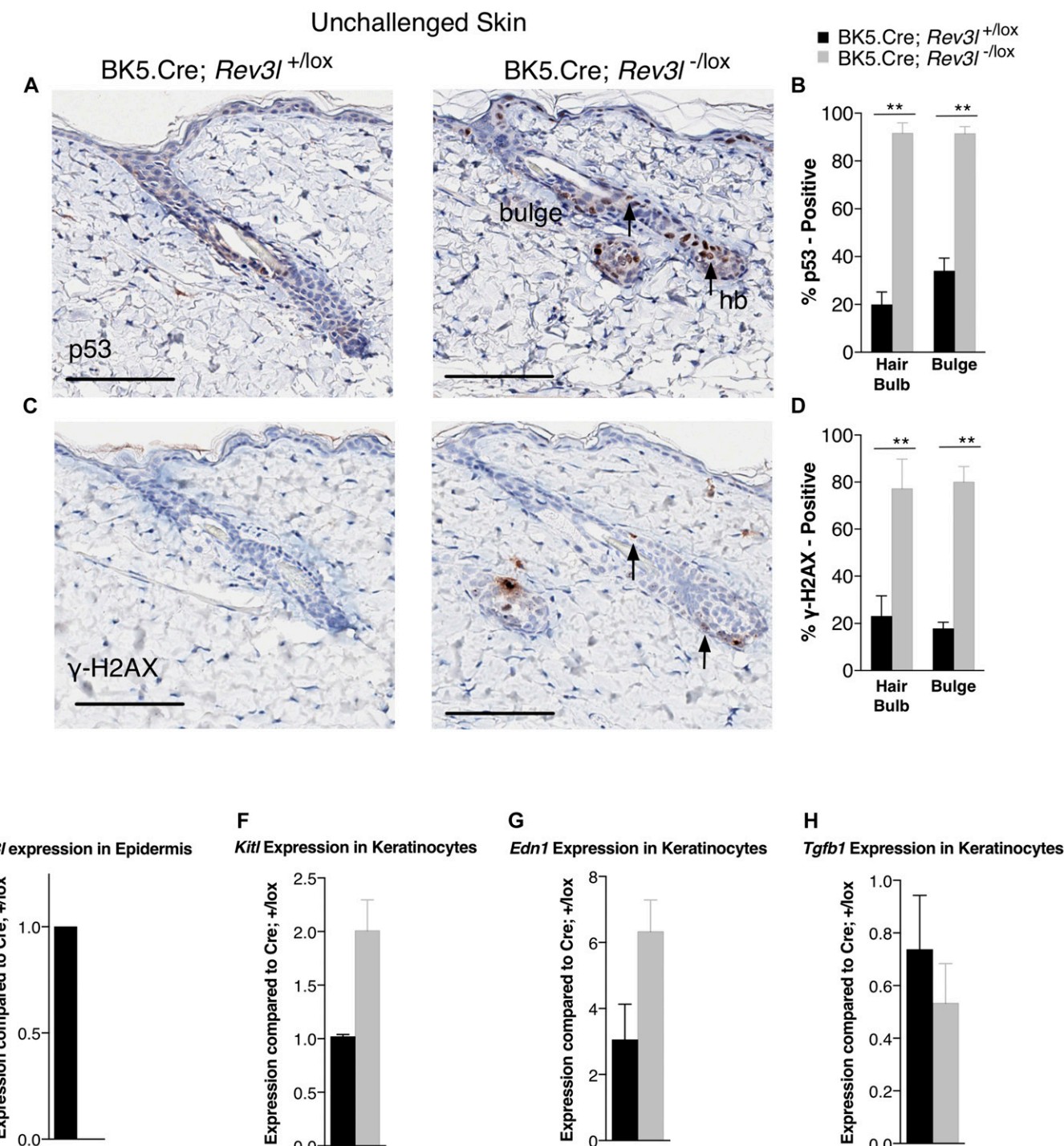

**Figure 6. Increased DNA breaks and p53 in undamaged *Rev3l*-deficient hair follicles and increased expression of *Kitl* in *Rev3l*-deficient keratinocytes.**
**(A)** p53 immunohistochemistry of skin sections from untreated BK5.Cre; *Rev3l*+/lox and BK5.Cre; *Rev3l*−/lox mice; arrows point to examples of p53-positive cells. **(B)** Quantification of the percent of p53-positive cells in bulge regions and hair bulbs (hb) (n = 5 mice of each genotype). **(C)** γ-H2AX immunohistochemistry of skin sections; arrows point to examples of γ-H2AX–positive cells. Distance bar: 100 μm. **(D)** Quantification of the percent of γ-H2AX–positive cells in bulge regions and hair bulbs (n = 5 mice of each genotype). Relative expression in keratinocytes derived from 7 to 10-wk-old skin from mice (n = 3 mice of each genotype) for the genes **(E)** *Rev3l*, **(F)** *Kitl*, **(G)** *Edn1*, and **(H)** *Tgfb1*. Data points are mean ± SEM; *$P < 0.05$, **$P < 0.01$.

follicle in *Rev3l*-deficient mice appears to promote a response of melanocytes to stress signals from epidermal keratinocytes. The nature of this stress signal is not clear but may involve the activation of the protein kinase C pathway (as is seen with TPA treatment) or the creation of a cytokine gradient caused by UV radiation–, wounding-, or TPA-induced epidermal inflammation.

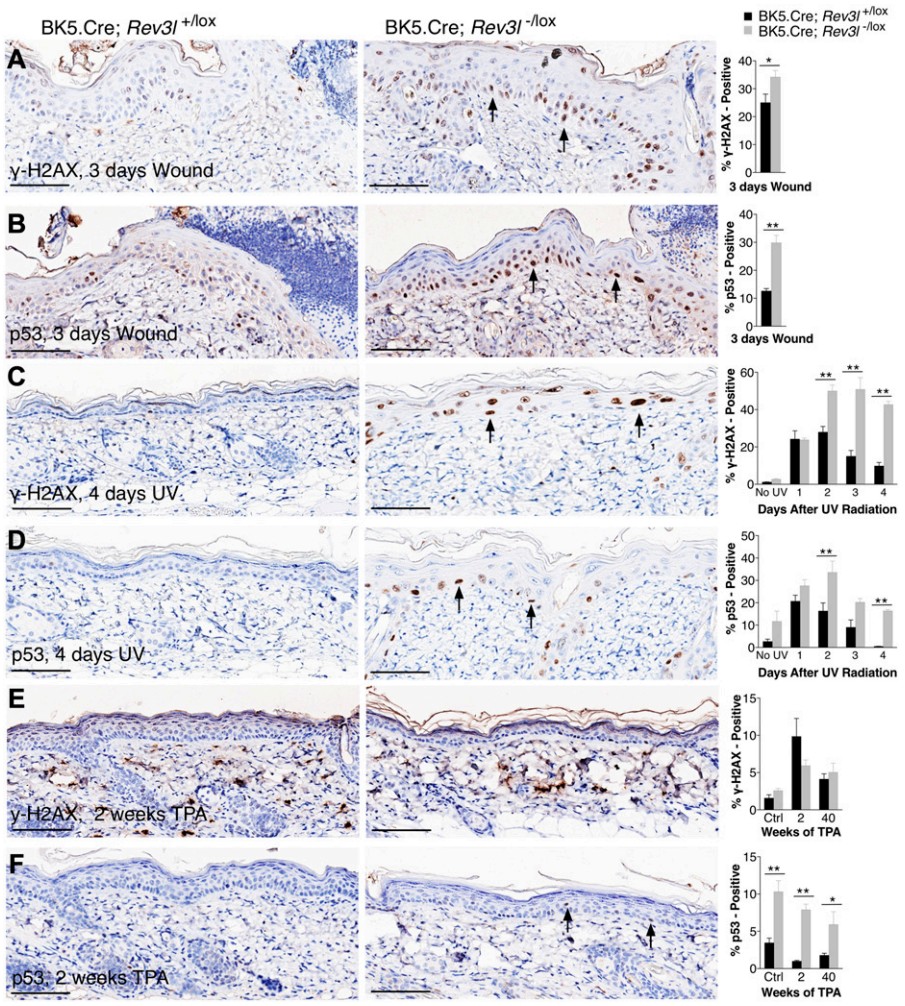

**Figure 7. Increased γ-H2AX and p53 stabilization in the epidermis of stressed *Rev3l*-deficient skin.**
γ-H2AX and p53 immunohistochemistry of skin sections from BK5.Cre; *Rev3l*+/lox and BK5.Cre; *Rev3l*−/lox mice **(A, B)** 3 d after wounding (the wound margin is shown), **(C, D)** 4 d after UV radiation, or **(E, F)** 2 wk after TPA application. Arrows point to examples of γ-H2AX– or p53-positive keratinocytes. Distance bar: 100 μm. The graphs show quantification of percent of γ-H2AX– or p53-positive cells in the epidermis of mice 3 d after wounding (n = 5 mice of each genotype); with no UV or 1–4 d after UV radiation (n = 4 mice of each genotype); or 2 wk after acetone or TPA application, or 40 wk after TPA application (n = 4 mice of each genotype). Data points are mean ± SEM; *P < 0.05, **P < 0.01.

The process of epidermal pigmentation in the haired skin of *Rev3l*-deficient mice appears to be p53 dependent. This is supported by the lack of melanocytes or pigmentation in p53-negative areas of UV-irradiated epidermis. The origin of these p53-negative areas is not known, but they are unlikely to be p53-mutant patches because the pAb240 antibody used in this study recognizes both wild-type and mutant conformations of p53. P53 is likely to be a key player in epidermal pigmentation because in keratinocytes P53 directly up-regulates the transcription of genes that encode proteins that are secreted and bind to the melanocortin 1 receptor on melanocytes. These include KITL (27), POMC (29), and EDN1 (30). Activation of melanocortin I receptor stimulates production of eumelanin, increases pigmentation, and may increase repair of common UV-induced DNA photoproducts (39). The activation occurs via palmitoylation by ZDHHC13, an enzyme with multiple substrates and pleiotropic effects in mouse skin (40).

We presented some evidence here that elements required for the pigmentation pathway are activated in the absence of pol ζ function. The KITL axis is up-regulated in *Rev3l*-deficient keratinocytes compared with *Rev3l*-proficient keratinocytes. The significant twofold difference is inclusive of the entire population of *Rev3l*-deficient skin cells and so probably underestimates the activation

seen in the target proliferating cells that have an activated p53 pathway.

## Relevance to human pathologies

It is important to understand pathways affecting skin pigmentation, given the roles of melanocytes in protecting against UV radiation damage and in the pathogenesis of melanoma in humans. The strong signaling invoked by a pol ζ defect is expected to be useful in models studying melanomagenesis, wound healing problems, and pigmentation abnormalities (including genetic syndromes with overactive KITL).

A summary of the consequences of pol ζ deficiency in keratinocytes is outlined in Fig 9. The most remarkable feature is the damage-induced migration of melanocytes to the epidermis in mice that have a disruption of *Rev3l* in the keratinocytes. The precise mechanism is unknown, but apparently occurs via signaling to *Rev3l*-disrupted hair follicles (possibly through a cytokine gradient). Keratinocytes in both the epidermis and the follicles are under baseline DNA damage stress as shown by DNA breaks and p53 activation. Epidermal wounding, or UV radiation, causes attempted proliferation of *Rev3l*-defective epidermal keratinocytes,

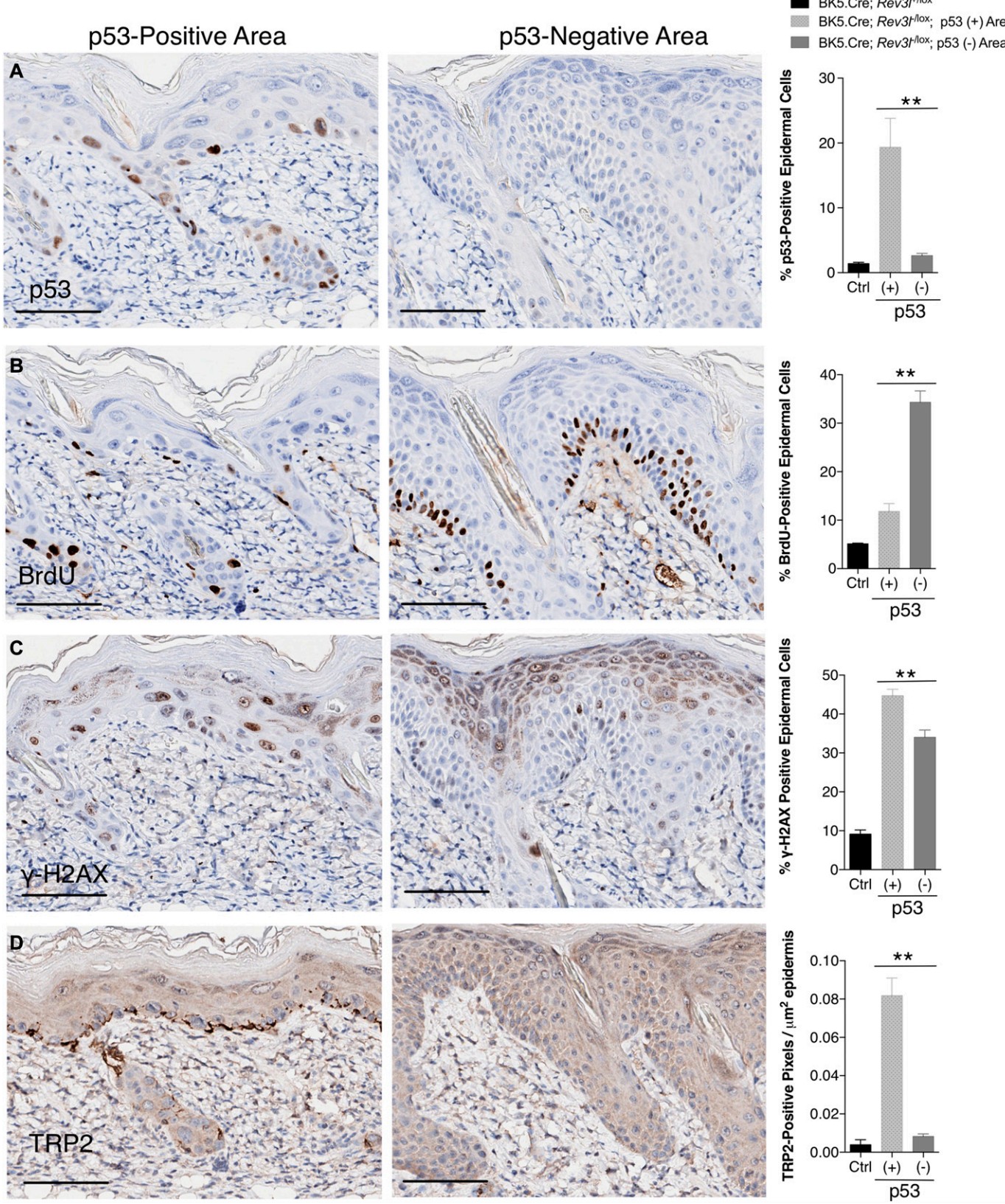

p53-Positive Area

p53-Negative Area

BK5.Cre; *Rev3l*⁺/lox
BK5.Cre; *Rev3l*⁻/lox; p53 (+) Area
BK5.Cre; *Rev3l*⁻/lox; p53 (-) Area

A  p53

B  BrdU

C  γ-H2AX

D  TRP2

which greatly increases DNA breaks and p53 activation. This generates a signal strong enough to cause the migration of melanocytes to the epidermis. Stabilized p53 activates the expression of pro-pigment genes, such as *Kitl* and *Pomc*, which produce proteins that signal to melanocytes. Once in the epidermis, melanocytes pigment the epidermal keratinocytes in response to DNA damage/stress signals.

## Materials and Methods

### Mouse strains and breeding

All animal work in this study were completed according to the University of Texas MD Anderson Cancer Center Institutional Animal Care and Use Committee guidelines. Experiments done on pigmented mice used animals on a C57BL6/J background with residual strain 129 background. Mice expressed the BK5.Cre transgene ([17]) and the mT/mG Cre reporter transgene ([16]), with either the *Rev3l* wild-type (+), knockout (–), or floxed (lox) alleles. All other experiments (non-pigmented UV radiation and TPA experiments) were carried out using tissues from the animals and experiments described ([14]). These mice were on an albino strain background (tyrosinase mutation) that was at least 95% FVB/N. The FVB strain mice that originate from the Taconic laboratory (FVB/T) have an additional mutation in *Skint1*, which leads to a defect in skin $\gamma-\delta$ T cells ([41]). We genotyped for this mutation and found that our FVB/N mice carried the wild-type allele of *Skint1* as expected.

### Ex vivo wound healing experiments

Ex vivo wound healing experiments were conducted as described ([42], [43]). 7–10-wk-old pigmented mT/mG mice with the genotypes BK5.Cre; *Rev3l*[+/lox] and BK5.Cre; *Rev3l*[–/lox] were euthanized by $CO_2$ asphyxiation followed by cervical dislocation, and their dorsal skin was removed. Twelve 4-mm punch biopsies (Accupunch; Xemax Surgical Products) from each mouse skin were taken and placed into a 24-well tissue culture plate. 200 $\mu$l explant growth medium/well was added to each well and the explants were placed into a 37°C, 5% $CO_2$-humidified incubator for 16 h. An additional 1.5 ml of explant growth medium was then added to each well. Over the next 4 d, the keratinocytes began to migrate out of the explants. Daily starting at day 5, the expanding explants were photographed using a Nikon TS-100 fluorescent microscope equipped with a DS-L2 camera at 40× magnification, detecting GFP (keratinocytes) and RFP (fibroblasts) fluorescence. Each picture had a scale bar added to it, and all the pictures from one explant (up to 18 pictures per explant) were merged using the Adobe Photoshop CS4 Photomerge function. The area of cell dispersion was measured from these merged pictures. In addition, at 5 d after plating, Hoechst 33342 dye (10 $\mu$g/ml) was added to six explants to label live cell nuclei, and

these were incubated for 1 h at 37°C. Pictures were then taken each of the GFP (keratinocytes) and Hoechst 33342 (blue) fluorescence on the Nikon fluorescent microscope at 100× magnification. These pictures were merged using the Photoshop Photomerge function, the area of outgrowth was measured, and then the number of Hoechst-stained nuclei were counted to obtain the number of cells per area of outgrowth. The average number of cells per area of outgrowth for each explant was multiplied by total outgrowth area to obtain the total number of cells per explant.

### In vivo wound healing experiments

7–10-wk-old pigmented mT/mG mice with the genotypes BK5.Cre; *Rev3l*[+/lox] and BK5.Cre; *Rev3l*[–/lox] were used. The dorsal skin of the mice was shaved and 24 h later the skin was depilated with Nair (Church & Dwight). The following day, the mice were anesthetized with isoflurane and a strip of siliconized tape (3M) was placed across their shoulder blades (this prevents wound retraction caused by contraction of the muscles under the skin). 5-mm wounds were made using an Accupunch through the tape. While under anesthesia, GFP fluorescence in the epidermis of these mice was imaged on an IVIS spectrum (using epifluorescence with filter set 491/501), which allowed us to examine the wound area and its changing size during the process of re-epithelialization. Pictures were taken of the GFP fluorescence on the IVIS spectrum every 24 h for 10 d. The wound area of these mice was measured over time using Adobe Photoshop. After the fluorescent pictures were taken on the 10th day, the mice were injected i.p. with BrdU (100 $\mu$g/g bodyweight) and 30 min later were killed by $CO_2$ asphyxiation followed by cervical dislocation. The dorsal skin was removed and fixed with 10% (wt/vol) formalin in PBS. After 24 h, the skin was moved to 70% (vol/vol) ethanol, and then was paraffin-embedded, sectioned, and stained. Another cohort of mice was wounded, as above, and then allowed to heal for 3 d before they were injected with BrdU and killed (no IVIS spectrum pictures were taken). The dorsal skin of these mice was also removed, and fixed, sectioned, and stained as above.

### Exposure of mice to UV radiation

The 7–10-wk-old pigmented mT/mG mice were shaved and their skin was depilated with Nair 48 h before UV radiation. The mice were put into single-animal Plexiglass containers in a carousel, so that each animal received an equivalent dose of UV radiation. They were irradiated with 1,200 J/m$^2$ UVB using Phillips UVB TL 20W/12 RS lights at a fluence of ~5 J/m$^2$·s; immediately afterward, their dorsal skin was photographed with a Canon EOS Rebel XSi digital SLR camera; and the mice were returned to their cages. Every 24 h for 10 d, the dorsal skin of the mice was photographed. On the 10th day, the mice were injected with BrdU (100 $\mu$g/g bodyweight) and 30 min later were killed by $CO_2$ asphyxiation, followed by cervical dislocation.

**Figure 8. Loss of p53 correlates with loss of melanocytes and increased proliferation in UV-irradiated, *Rev3l*-deficient epidermis.**
p53, BrdU, TRP2, and $\gamma$-H2AX immunohistochemistry of p53-positive and p53-negative sections of epidermis from BK5.Cre; *Rev3l*[–/lox] mice 5 d after UV radiation. **(A)** p53-positive and p53-negative regions. **(B)** BrdU-positive cells in these regions. **(C)** $\gamma$-H2AX–positive cells. **(D)** Melanocytes (TRP2-positive cells), pixels/$\mu$m$^2$ epidermis, in p53-positive or p53-negative epidermis from BK5.Cre; *Rev3l*[–/lox] mice (n = 5 mice of each) 5 d after UV radiation, compared with control BK5.Cre; *Rev3l*[+/lox] skin at the same time point. Data points are mean ± SEM; *$P < 0.05$, **$P < 0.01$. Distance bar: 100 $\mu$m.

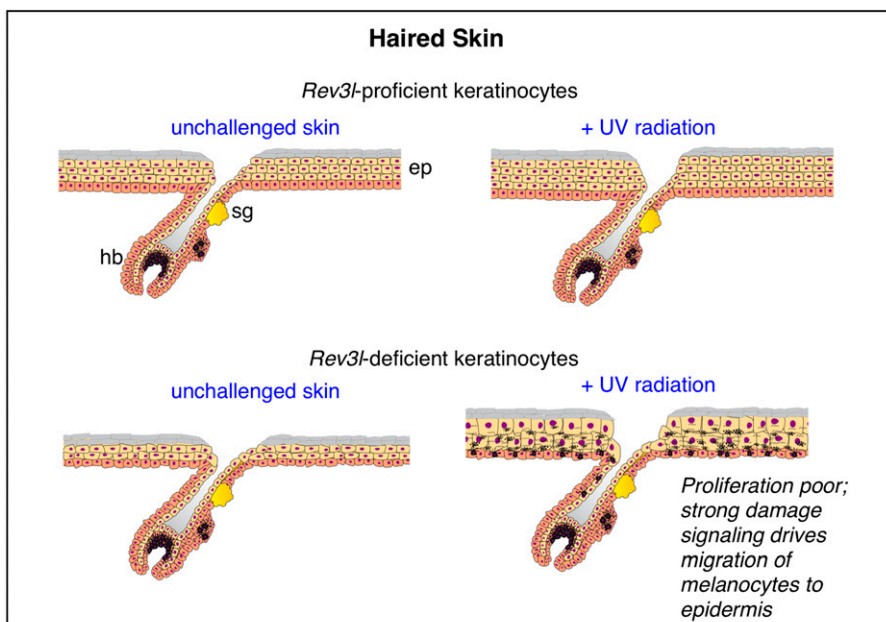

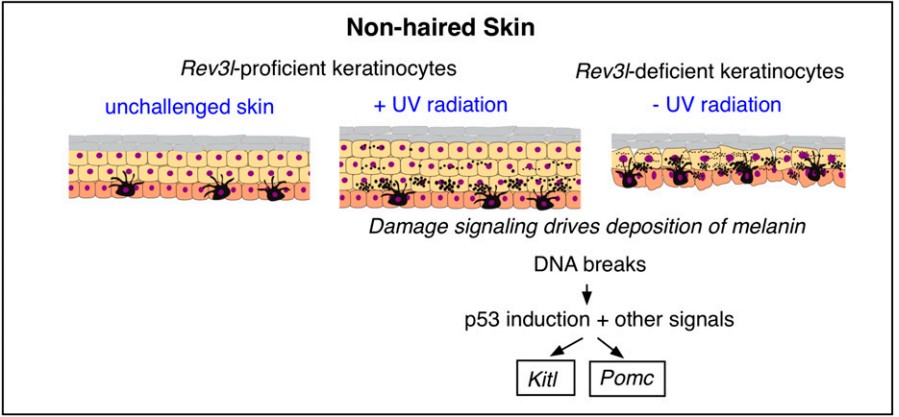

**Figure 9. Summary of defective wound healing and hyperpigmentation-dependent on Pol ζ.**
In haired skin (top panel), mice expressing *Rev3l* in epidermal keratinocytes can proliferate in response to UV radiation, but melanocytes are not mobilized from the hair follicles. In *Rev3l*-defective epidermis, p53 and other stress signals are activated at a high level, including some that signal to melanocytes (*Kitl* and *Pomc*). Damaged pol ζ–defective keratinocytes proliferate poorly in response to wounding or UV radiation. Strong damage-initiated signaling drives migration of melanocytes to epidermis and activation of melanin secretion. Non-haired mouse skin (lower panel), as in the feet and the ears, harbors epidermal melanocytes. Unchallenged skin with *Rev3l*-defective keratinocytes is under constitutive stress sufficient to deposit increased melanocytes in the epidermis. A stronger stress signal engendered by wounding or UV radiation induces migration of melanocytes from the hair follicles to the epidermis, with deposition of melanin. sg, sebaceous gland; hb, hair bulb; ep, epidermis.

The dorsal skin of the mice was again photographed and then was removed and fixed with 10% (wt/vol) formalin in PBS. After 24 h, the skin was moved to 70% (vol/vol) ethanol; it was then paraffin-embedded, sectioned, and stained. All of the experiments on UV-irradiated non-pigmented mice used tissues mentioned in reference 14. Briefly, these albino FVB/N mice were treated with 1,800 J/$m^2$ UVB radiation and killed at 0, 1, 2, 3, 4, 5, 8, or 10 d after irradiation. Their dorsal skin was removed, fixed, and stained as described here.

### Exposure of mice to TPA

TPA (Sigma-Aldrich) was dissolved into acetone to a concentration of 3.4 nmol/200 μl. The dorsal skin of pigmented 7–10-wk-old mT/mG mice was shaved and 48 h later, 200 μl of TPA was applied to the dorsal skin. This was performed two times per week for 2 wk. 48 h after the final TPA application, the mice were injected with BrdU (100 μg/g bodyweight) and 30 min later were killed by $CO_2$ asphyxiation followed by cervical dislocation. The dorsal skin of the mice was shaved, depilated, and photographed with the Canon

digital camera; it was then removed and fixed with 10% (wt/vol) formalin in PBS. After 24 h, the skin was moved to 70% (vol/vol) EtOH; it was then paraffin-embedded, sectioned, and stained. All of the experiments on TPA-treated non-pigmented mice used tissues mentioned in reference 14. Briefly, these albino FVB/N mice were treated twice per week for 2 wk with vehicle only (acetone) or with 3.4 nmol TPA twice per week for 2 or 40 wk, and were killed 48 h after the final application. Their dorsal skin was removed, fixed, and stained as described here.

### Animal tissue histology and immunohistochemistry

Fixed skin tissues were embedded in paraffin and sectioned. The sections were stained with hematoxylin and eosin, hematoxylin alone, or with hematoxylin combined with the following antibodies: BrdU, total p53, and γ-H2AX (14); keratin-6 (rabbit polyclonal, 1:500; Covance); keratin-10 (rabbit polyclonal, 1:500; Covance); or TRP2 (rabbit polyclonal, 1:500; Abcam). The tissues were then labeled with secondary antibodies.EnVision+ System HRP labeled polymer;

anti-rabbit HRP (Dako) was used for the rabbit polyclonal antibodies. Melanin was stained with a Fontana-Masson stain kit (Sigma-Aldrich procedure no.: HT200). Stained slides were digitally scanned using a ScanScope CS (Aperio).

### Quantification of immunohistochemical staining of mouse tissues

Using the digital images of the slides, skin sections were chosen and antibody-positive cells were quantified from BrdU-, p53-, or $\gamma$-H2AX–stained slides (14). For the keratin-10 stain, the healed wound areas in the 10-d samples were selected in the digital slide. From there, the total epidermal cell number was counted and then was compared with the total cells that were epidermal but not staining for K10 (basal epidermal cells). Cell size was quantified by measuring total epidermal cell area (using the ImageScope program) and then dividing by the total number of epidermal cells in that area. TRP2-positive cells were quantified by selecting 11 scattered 1-mm sections of skin from the digital images, then counting the number of TRP2-positive epidermal cells in that 1-mm section. In the experiments quantifying the hyper-proliferative and control areas of skin, TRP2 was quantified by using the ImageScope-positive pixel counter and then comparing this with the total epidermal area. This was done because the epidermal area could be different between the hyper-proliferative and control skin. Hair follicles with p53- or $\gamma$-H2AX–positive cells in their bulge or hair bulb regions were also quantified. The bulge region was defined as the area of hair follicle a short distance below the sebaceous gland. The hair bulb is the region of the hair follicle immediately adjacent to the dermal papillae.

### Isolation and gene expression of keratinocytes

Pigmented 7–10-wk-old mT/mG mice were shaved and their skin was depilated 48 h before sacrifice. The animals were killed by $CO_2$ asphyxiation followed by cervical dislocation, and then the dorsal skin was removed. It was processed into a single-cell suspension by digestion with collagenase (Sigma-Aldrich), dispase (Sigma-Aldrich), and trypsin (Thermo Fisher Scientific) as in reference 44. Briefly, the skin was minced, washed with PBS, and incubated for 12–13 h in a collagenase/dispase solution. The cells were further incubated for 5 min with trypsin, followed by mechanical disruption using 18-, 20-, and 22-gage needles. The cells were incubated for 5 min with red blood cell lysis buffer (BioVision) and then washed and flow-sorted for RFP-positive (dermal) cells or GFP-positive (epidermal and hair follicle keratinocyte) cells. RNA was extracted from these cells and gene expression was assayed using an Applied Biosystems 7900HT Fast Real-Time PCR System. We measured expression of murine *Rev3l* (Ex26Fwd: 5′-GTG AAT GAT ACC AAG AAA TGG GG-3′; Ex27Rev: 5′-GTG AAT GAT ACC AAG AAA TGG GG-3′; Probe: FAM-MGB-5′-TAC TGA CAG TAT GTT TGT-3′), *Trp53*, *Kitl*, *Pomc*, *Wnt1*, *Edn1*, and *Tgfbr1* (all reactions except *Rev3l* were premade assays from IDT). Relative expression was achieved by comparing the ΔCt values of each sample with the ΔCt from one BK5.Cre; *Rev3l*$^{+/lox}$ mouse sample.

### Statistical analysis

Analysis of the outgrowth area or cell number of the explants, or the TRP2-, p53-, or $\gamma$-H2AX–positive cells in UV-irradiated or TPA-treated skin by immunohistochemistry was performed using a two-way ANOVA ($P < 0.05$). Statistical analysis of the immunohistochemical staining of the wounds or undamaged hair follicles or the gene expression analysis was performed by unpaired $t$ test. Comparison of immunohistochemical staining in the hyper-proliferative, p53-positive and p53-negative epidermal areas was performed using one-way ANOVA with Tukey's multiple comparisons test ($P < 0.05$).

## Supplementary Information

## Acknowledgements

We thank Megan Lowery for assistance with figures. We appreciate helpful discussions and advice from MD Anderson Cancer Center colleagues Junya Tomida, David Johnson, Fernando Benavides, and Carlos Perez. We appreciate assistance from the MD Anderson Research Animal Support Facility and the Research Histology, Pathology and Imaging Core. These studies were funded by National Institutes of Health P01 grant CA193124 (RD Wood) and the Grady F. Saunders PhD Distinguished Research Professorship (RD Wood). The Research Animal Support Facility and the Research Histology, Pathology and Imaging Core were supported by MD Anderson Cancer Center Support grant P30-CA016672 from the National Institutes of Health.

### Author Contributions

SS Lange: conceptualization, investigation, methodology, and writing—original draft.
S Bhetawal: conceptualization, investigation, methodology, and writing—review and editing.
S Reh: investigation and writing—review and editing.
KL Powell: investigation.
DF Kusewitt: conceptualization, investigation, methodology, and writing—review and editing.
RD Wood: conceptualization, supervision, funding acquisition, and writing—review and editing.

### Conflict of Interest Statement

The authors declare that they have no conflict of interest.

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
