## [Reviewer comments · Life Science Alliance]

DNA polymerase ζ -deficiency causes impaired wound healing and stress-induced skin pigmentation

Sabine S. Lange, Sarita Bhetawal, Shelley Reh, K. Leslie Powell, Donna F. Kusewitt, and Richard D. Wood

DOI: 10.26508/lsa.201800048

Review timeline:

Submission Date:	7 March 2018
Editorial Decision:	11 April 2018
Revision Received:	14 June 2018
Editorial Decision:	15 June 2018
Accepted:	20 June 2018

Report:

(Note: Letters and reports are not edited. The original formatting of letters and referee reports may not be reflected in this compilation.)

1st Editorial Decision

11 April 2018

Thank you for submitting your manuscript entitled "DNA polymerase ζ -deficiency causes impaired wound healing and stress-induced skin pigmentation" to Life Science Alliance. The manuscript was assessed by expert reviewers, whose comments are appended to this letter. We invite you to submit a revision if you can address the reviewers' key concerns, as outlined here.

As you will see, the reviewers appreciate your data and provide constructive input on how to revise your manuscript to make it suitable for publication in Life Science Alliance. The controls/repeats requested by reviewer #1 should be provided, and alternative hypotheses should get discussed in absence of further genetic analyses (reviewer #2). The revision seems minor and straightforward in our view, please let us know in case you would like to discuss individual points further.

Thank you for this interesting contribution to Life Science Alliance. We are looking forward to receiving your revised manuscript.

REFeree REPORTS

Reviewer #1 (Comments to the Authors (Required)):

In the manuscript entitled "DNA polymerase ζ -deficiency causes impaired wound healing and stress-induced skin pigmentation", Lange and colleagues discovered that disruption of Rev3l the catalytic subunit of pol ζ Rev3l in mouse epidermis leads a defect of cutaneous wound healing. Overall the story is interesting, but a few technique concerns need to be addressed:

1. Fig 1c is unclear, cannot see the shape of the keratinocytes, it is hard to drudge whether these green staining are non-specific signals.
2. For Figure 3, the authors should try Fontana-Masson staining, which is a widely used melanin stain method to confirm the result.
3. For Figure 4, the quality of the staining is poor, the result cannot confirm melanocytes are in the epidermis. Because melanocytes are also in the hair follicle, the authors should at least show positive melanocytes staining in hair follicles in both mice, and positive melanocytes staining in epidermis in $-lox$ mice.

Reviewer #2 (Comments to the Authors (Required)):

In this manuscript, Sabine Lange and colleagues study the potential role of DNA polymerase zeta in cell proliferation and wound healing. By observing wound healing in Rev3-proficient and deficient epidermis, authors show Rev3 deficiency leads to reduced cell growth and thus abnormal wound healing. Interestingly, the Rev3 deficiency is associated with increased pigmentation in the skin following wound healing. The increased pigmentation can also be induced by UV radiation and TPA treatment. Authors further show that P53 is activated in Rev3-deficient cells and is required for recruitment of melanocytes and pigmentation. This study is nicely executed, and data well presented. The main concern here is that authors strongly favor the model that a P53 dependent signaling, possibly through the secreted proteins such as KITL, promotes melanocyte relocation and pigmentation in the skin. However, there is no direct evidence supporting the claim. Without evidences such as the knockout experiments of KITL in Rev3- cells, the model presented here is no more than speculation. Thus, authors should at least discuss other possibilities. For example, inactivation of P53 in Rev3-deficient cells simply allows for more robust cell growth and wound healing which alleviates the necessity of the cell migration and pigmentation.

Reviewer #3 (Comments to the Authors (Required)):

In this MS, the authors explored the impact of the disruption of Rev3L (encoding the catalytic subunit of the TLS polymerase zeta) in mouse epidermis. After introducing excisional wounds, they observed a markedly reduced proliferation of epidermal cells that had migrated into the wound area and, as a consequence, an abnormal wound architecture in Rev3L-deficient skin. Intriguingly, this defect is associated to an aberrant epidermal pigmentation due to a migration of melanocytes (that are Rev3L-proficient) to the epidermis of stressed Rev3L-deficient skin. They also showed that wounding induces the stabilization p53 and an accumulation of γ -H2AX foci in Rev3L-deficient epidermis and hair follicles. Interestingly, loss of p53 in Rev3L-deficient skin abolishes melanocytes recruitment to the wound areas. Thus the pigmentation response to proliferative stress in Rev3L-deficient skin is dependent on p53 stabilization.

This is a very well-written and well-organized paper. The figures are convincing and of high quality. The conclusions are supported by the analysis of the data presented. Although descriptive, this MS presents the results of original research and makes a valuable contribution to knowledge on the consequences of pol zeta deficiency. Therefore the paper can be accepted for publication in LSA.

1st Revision – authors' response

14 June 2018

Reviewer #1

In the manuscript entitled "DNA polymerase ζ -deficiency causes impaired wound healing and stress-induced skin pigmentation", Lange and colleagues discovered that disruption of Rev3l the catalytic subunit of pol ζ Rev3l in mouse epidermis leads a defect of cutaneous wound healing. Overall the story is interesting, but a few technique concerns need to be addressed:

1. Fig 1c is unclear, cannot see the shape of the keratinocytes, it is hard to drudge whether these green staining are non-specific signals.

Agreed; the pdf provided for review was low resolution. A new image has been substituted, which is better. It shows that only the epidermis and hair follicles have green staining. To detect the GFP staining, the samples were not formalin fixed, and so the morphology is not quite as defined as in the rest of the paper. This panel is for illustration of the cre action; other examples from this mouse model are shown in our reference 14 (Lange et al 2013 PNAS)

2. For Figure 3, the authors should try Fontana-Masson staining, which is a widely used melanin stain method to confirm the result.

Thank you for this suggestion. The new Figure 3 depicts the Fontana-Masson staining, which worked very well. The hematoxylin-only staining is shown in a new Figure S1.

3. For Figure 4, the quality of the staining is poor, the result cannot confirm melanocytes are in the epidermis. Because melanocytes are also in the hair follicle, the authors should at least show positive melanocytes staining in hair follicles in both mice, and positive melanocytes staining in epidermis in -/lox mice.

We have provided larger and higher resolution views in a new Figure 5 (former Figure 4), to show the TRP2 staining more clearly. A new simplified Figure S2 is provided, which shows positive melanocyte staining in hair follicles of both genotypes of mice. Fig 8D has related information.

Reviewer #2:

In this manuscript, Sabine Lange and colleagues study the potential role of DNA polymerase zeta in cell proliferation and wound healing. By observing wound healing in Rev3-proficient and deficient epidermis, authors show Rev3 deficiency leads to reduced cell growth and thus abnormal wound healing. Interestingly, the Rev3 deficiency is associated with increased pigmentation in the skin following wound healing. The increased pigmentation can also be induced by UV radiation and TPA treatment. Authors further show that P53 is activated in Rev3-deficient cells and is required for recruitment of melanocytes and pigmentation. This study is nicely executed, and data well presented. The main concern here is that authors strongly favor the model that a P53 dependent signaling, possibly through the secreted proteins such as KITL, promotes melanocyte relocation and pigmentation in the skin. However, there is no direct evidence supporting the claim. Without evidences such as the knockout experiments of KITL in Rev3- cells, the model presented here is no more than speculation. Thus, authors should at least discuss other possibilities. For example, inactivation of P53 in Rev3-deficient cells simply allows for more robust cell growth and wound healing which alleviates the necessity of the cell migration and pigmentation.

Thank you; we have modified the discussion on pages 11 and 12 of the manuscript to indicate that our model is a suggested one, and there may be multiple factors behind the increased pigmentation. Some of this thinking is illustrated in Fig 9, where the importance of the MSH/Pomc pathway is indicated as well as p53-dependent and independent pathways. As summarized in the discussion, multiple pieces of evidence support a p53-mediated mechanism, but we agree that activation of p53 (or a p53 defect) will have multiple consequences. Inactivation only of p53 does not seem to rescue a pol zeta defect, as we and others have explored – our reference 11, (Wittschieben et al 2010 PNAS).

Reviewer #3:

In this MS, the authors explored the impact of the disruption of Rev3L (encoding the catalytic subunit of the TLS polymerase zeta) in mouse epidermis. After introducing excisional wounds, they observed a markedly reduced proliferation of epidermal cells that had migrated into the wound area and, as a consequence, an abnormal wound architecture in Rev3L-deficient skin. Intriguingly, this defect is associated to an aberrant epidermal pigmentation due to a migration of melanocytes (that are Rev3L-proficient) to the epidermis of stressed Rev3L-deficient skin. They also showed that wounding induces the stabilization p53 and an accumulation of γ -H2AX foci in Rev3L-deficient epidermis and hair follicles. Interestingly, loss of p53 in Rev3L-deficient skin abolishes melanocytes recruitment to the wound areas. Thus the pigmentation response to proliferative stress in Rev3L-deficient skin is dependent on p53 stabilization.

This is a very well-written and well-organized paper. The figures are convincing and of high quality. The conclusions are supported by the analysis of the data presented. Although descriptive, this MS presents the results of original research and makes a valuable contribution to knowledge on the consequences of pol zeta deficiency. Therefore the paper can be accepted for publication in LSA.

2nd Editorial Decision

15 June 2018

Thank you for submitting your revised manuscript entitled "DNA polymerase ζ -deficiency causes impaired wound healing and stress-induced skin pigmentation".

We appreciate the introduced changes, and as you will see below, reviewer #1 also thinks that her/his concerns have been satisfactorily addressed. I am therefore happy to accept your manuscript in principle for publication in Life Science Alliance. Congratulations on this very nice work.

REFEREE REPORTS

Reviewer #1 (Comments to the Authors (Required)):

The authors have addressed all my concerns.